



# Pathways to bring the costs down of floating offshore wind farms in the Atlantic Area

Juan José Cartelle-Barros[1], David Cordal-Iglesias[2], Eugenio Baita-Saavedra[2], Almudena Filgueira-Vizoso[1], Bernardino Couñago-Lorenzo[3], Fernando Vigara[3], Carlos Cortés[3], Lara Cerdán[3], Javier Nieto[3] ,
José Serna[3] and Laura Castro-Santos[4]

[1]Universidade da Coruña, Departamento de Química, Escola Politécnica Superior, 15471 Ferrol, Spain
[2]Universidade da Coruña, Escola Politécnica Superior, 15471 Ferrol, Spain
[3]Esteyco, SA, Menéndez Pidal, 17, 28036 Madrid, Spain
[4]Universidade da Coruña, Departamento de Enxeñaría Naval e Industrial, Escola Politécnica Superior, 15471 Ferrol, Spain

*Correspondence to*: Laura Castro-Santos (laura.castro.santos@udc.es)

**Abstract.** Every nations' development lies on the electricity production, since it facilitates life and development of their society (heating, lighting, etc.). Nevertheless, conventional power plants, which use fossil fuels, cause environmental impacts, such as
global warming, acidification, eutrophication, among many others. In addition, these conventional resources generate a dependence of external providers, which obstructs the progress of the developing countries. Renewable energies came to solve part of these problems. In this context, wind energy is one the technologies with more expansion all over the world. Offshore locations have a better wind resource than onshore ones and their exploitation is lower. The objective of this work is to present a holistic approach to assess the feasibility of floating offshore wind farm in a life cycle perspective. The methodology
proposed analyses the Net Present Value, the Internal Rate of Return, the Payback Period and the Levelized Cost of Energy of the farm. The case study is built based on a disruptive floating spar-type platform called TELWIND®, to be implemented in the Atlantic Area region. Results indicate how important these parameters are in economic terms and show the pathways to reduce the costs of this type of infrastructures Furthermore, the methodology proposed allows the selection of the best region where a floating offshore wind farm can be installed. Finally, this study can be useful for Governments and relevant authorities
to determine the best location of a floating offshore wind farm and develop the roadmap of offshore wind in their country.

## 1 Introduction

In the global context, the use of renewable energy is becoming very important due to the need to reduce greenhouse gases (Enkvist et al., n.d.). In the Paris agreement (Union, n.d.) it was established as a priority objective to reduce greenhouse gases
by 20% with respect to the year 1990 and reach 80% by the year 2050 (2009/28/EC, 2002). The percentage of energy from renewable sources in the final gross energy consumption has doubled from 8.5% in 2004 to 17% in 2016 (Eurosat Statistics Explained, n.d.). Within the map of renewable energies one of the most important is wind energy. In 2019, a total net capacity





of 189GW (Wind Europe, n.d.) and 591GW worldwide was installed in Europe in 2019 according to (Viaintermedia.com, n.d.). Spain is the fourth country with the highest installed wind power capacity, with 23 GW covering 18% of Spain's

electricity supply (Anon, n.d.).

Within wind energy, offshore wind energy is reaching very important values, changing the energy balance in favor of this last one. Analyzing wind power, it can be seen that offshore wind power has increased by 19.7% in the last 10 years (Wind Europe, n.d.),  being therefore one of the renewable energies with the greatest potential (Sun et al., 2012).

Its commercial use began in 2008 and from this moment the technology of each of its components is being optimized in order

to make it more viable both technically and economically. The first units were installed in the United Kingdom and Portugal (Seagen and Pelamis, respectively) (Esteban and Leary, 2012). The improvement of technology contributes to reduce costs and therefore facilitating its economic viability, thus causing the policies of different countries to be more likely to install this type of energy.

Within the offshore wind energy there are two main types: fixed (up to 50 meters deep) and floating (over 50 meters). This

article will analyse the floating equipment. Within the floating ones, the only commercial options are made of steel, but currently floating concrete platforms are being studied, since this material has characteristics that make it favorable for deep areas. Specifically, in this article, a 10MW TELWIND® concrete platform is addressed. The study is based on analyzing the costs associated with each of the parts involved in the process such as: conception and definition, design and development, manufacturing, installation and commissioning, exploitation and dismantling in different locations in Spain.

The cost of the life cycle will be considered (Castro-Santos and Diaz-Casas, 2014)(Scheu et al., 2012)(Topham and McMillan, 2017) to determine the main economic parameters that will determine the economic viability of the park: life cycle cost, Levelized Cost Of Energy (LCOE), Internal rate of return (IRR) and net present value (NPV). On the other hand, it is important to take into account distances to the coast, depths, restricted fishing or maritime traffic areas, etc. (Reimers et al., 2014)

The aim of this work is to present a holistic approach to assess the feasibility of a floating offshore wind farms in a life

cycle perspective. The methodology proposed analyses NPV, IRR and LCOE of the farm. The case study is built based on a disruptive floating spar-type platform called TELWIND, to be implemented in the Atlantic Area region. Results indicate a preliminary vision of how important these parameters are in economic terms and show the pathways to reduce the costs of this type of infrastructures Furthermore, the methodology proposed allows the selection of the best region where a floating offshore wind farm can be installed.

**2 Method**

Next, the calculation procedure will be explained, which is divided into the wind production calculation, the cost analysis and the obtaining of the economic parameters. A very important factor of this study is that the material of the platform concept is concrete, and therefore the references used to define the methodology will be modified to apply them to this case.



## 2.1 Calculation of wind energy generated

To obtain annual wind energy production at a given location, two factors must be taken into account: the wind turbine power curve and the wind distribution function.

In order to characterize the wind at each study location, the Weibull distribution function has been used. With this function it is possible to analyse the variation of speeds in a specific location through parameters that define its probability function.

$$f(v) = \left(\frac{k}{c}\right) \cdot \left(\frac{v}{c}\right)^{k-1} \cdot e^{\left[-\left(\frac{v}{c}\right)\right]^k} \tag{1}$$

where:

k: shape factor of the Weibull distribution.

c: scale factor of the Weibull distribution.

v: wind velocity (m/s).

The revenues from the offshore wind farm depend directly on energy production, and this will be greater as the wind speed increases up to the speed out of the wind turbine. It is very important to have reliable data and as accurate as possible,

considering that income is one of the most important factors in the economic feasibility assessment.

## 2.2. Calculation of the Life cycle cost

Understanding the life cycle is necessary to study the costs of a floating offshore wind farm and, thus, to perform the economic analysis based on the results obtained. This part of the study is called analysis of the Life-cycle Cost System (LCS).

The LCS of a floating offshore wind farm consists of several phases, which have been named by C1, C2, C3, etc., and

correspond to a stage of the life cycle. These stages range from the conception and definition of the wind farm (C1) to its final dismantling (C6), through the design and development (C2), manufacturing (C3), installation and commissioning (C4) and exploitation (C5).

The total cost system of the life cycle of a floating offshore wind farm (LCS) is obtained from (2).

$$LCS = C_1 + C_2 + C_3 + C_4 + C_5 + C_6 \tag{2}$$

In each of the phases several equations have been used (see Figure 1). These equations have been fed with a total of 938 inputs,

which are usually obtained from other stages of the project due to their dependency on the study location and the technology used (bathymetry, period and height of waves, characteristics of the platforms, wind speed, etc).





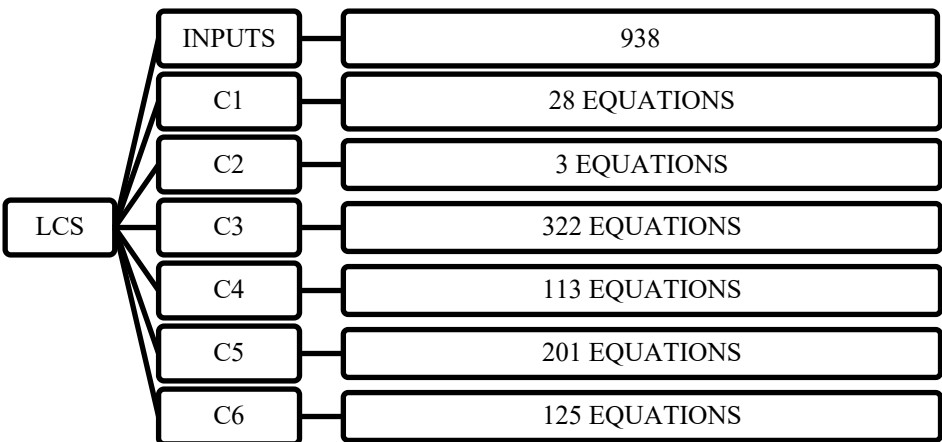

**Figure 1: LCS phases equations and inputs. Source:** (Castro-Santos and Diaz-Casas, 2014)**.**

Once LCS has been completely analysed and all inputs and information of the locations obtained, the economic parameters
can be calculated (see Figure 2).

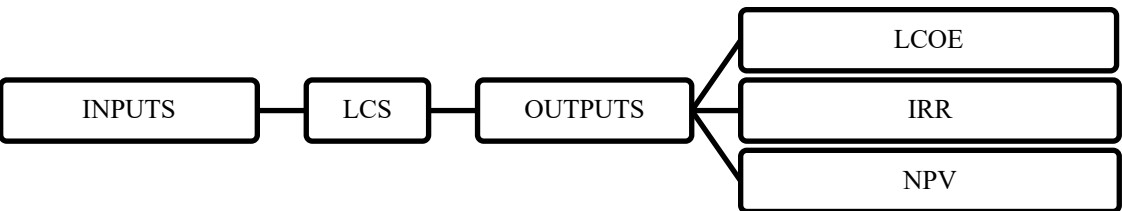

**Figure 2: Methodology scheme. Source:** (Castro-Santos and Diaz-Casas, 2014)**.**

### 2.3 Calculation of the economic parameters

Economic parameters are used to identify the economic viability of one or more projects in order to determine the investment
decision making. The most commonly used economic parameters are the net present value (NPV) and the internal rate of return
(IRR). In addition, for this type of comparative study it is interesting to know the levelized cost of energy (LCOE).

Figure 3 shows the equations of the economic parameters. The NPV allows us to calculate the present value of a certain number
of future cash flows ($CF_n$) that have originated or are the result of a specific investment.



**Net Present Value**

$$NPV = -CF_0 + \frac{CF_1}{1+r} + \frac{CF_2}{1+r} + \cdots + \frac{CF_{n-1}}{(1+r)^{n-1}} + \frac{CF_n}{(1+r)^n}$$

**Internal Rate of Return**

$$TIR \rightarrow 0 = -CF_0 + \frac{CF_1}{1+r} + \frac{CF_2}{1+r} + \cdots + \frac{CF_{n-1}}{(1+r)^{n-1}} + \frac{CF_n}{(1+r)^n}$$

**Levelized Cost of Energy**

$$LCOE = \frac{a \cdot INV \cdot OPEX}{NHA}$$


**Figure 3: Economic parameters equations.**

The higher the NPV of a project, the better results will give the necessary investment to carry it out. The IRR corresponds to the discount rate (r) that the NPV makes zero.

LCOE is a good indicator to compare different technologies, case studies or locations. The inputs that have been estimated to

calculate the LCOE are the initial investment (INV) and the operation and maintenance costs (OPEX). Annual energy production at location, obtained from the wind resource study, is also necessary to calculate the equivalent annual hours (NHA).

**3 Case of study**

This paper analyses an offshore wind farm of 200 MW composed of the floating offshore concrete platforms called TELWIND® (see **Figure 4**), designed by Esteyco. This platform is designed for a bathymetry of 110 m and it has an offshore

wind turbine of 10 MW. Its metacentric height inplace is higher than 3 m and its metacentric height transport is higher than 2 m. The overall heave period is higher than 30s and the overall pitch period is higher than 30s.



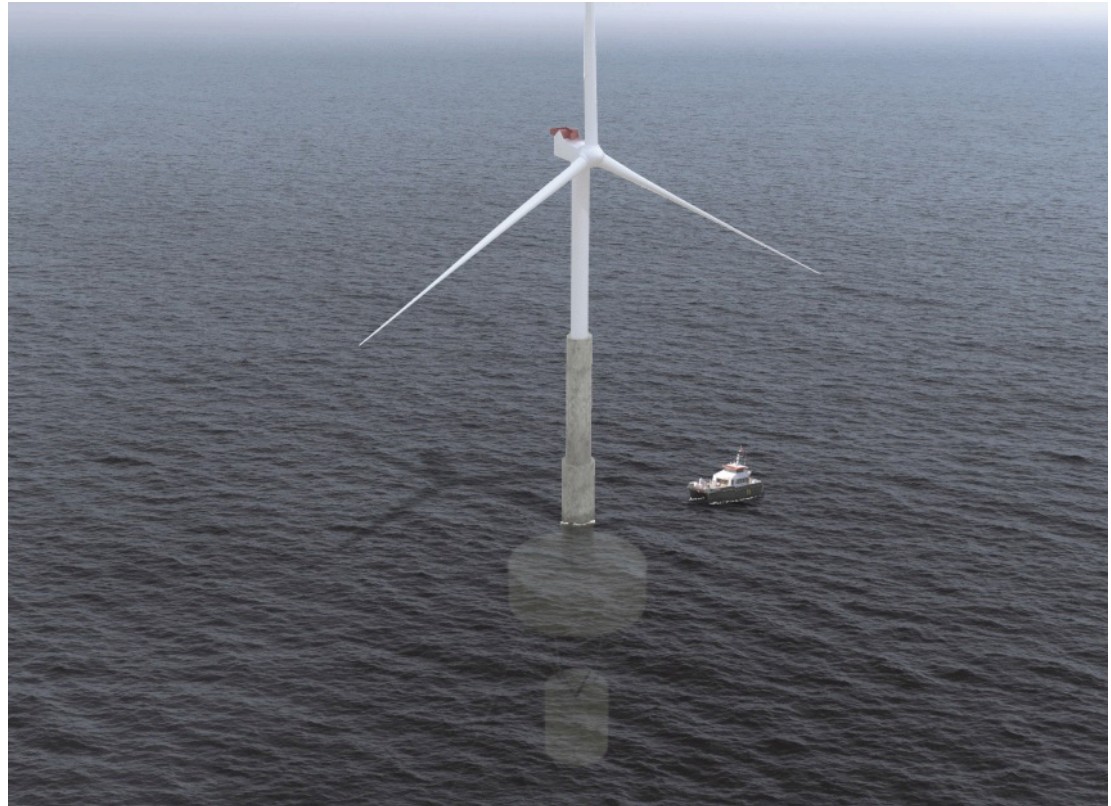

**Figure 4.** TELWIND® platform. Source: Figure courtesy of Esteyco [22]

The regions selected for this study are located in the Atlantic and Cantabria region of Spain: A Guarda-Baiona 1 (41.86 N, 9.18 W), A Guarda-Baiona 2 (41.86 N, 9.32 W), Ribadeo (43.83 N, 7.33 W), Navia (43.63 N, 6.53 W), San Vicente de la Barquera (43.56 N, 4.22 W), Santander (43.57 N, 3.66 W), Bilbao (43.67 N, 3.00 W), Mutriku (43.39 N, 2.33 W) and Huelva (36.76 N, 7.30 W).

Finally, three electric tariffs have been analyzed: 50 €/MWh (alternative 1), 100 €/MWh (alternative 2) and 150 €/MWh (alternative 3).

## 4 Results

By entering all the input data into the equations, the economic parameters of each location and alternative have been obtained, so that it was possible to compare each other and to obtain a series of conclusions in this regard.

**Figure 5** shows the life cycle costs of each proposed location considering dismantling costs (alternative 3), which helps us to identify the life cycle phases that have the greatest economic impact on an offshore wind farm at each study site. The costs of each location vary depending on the input data. A greater distance to the coast increases both transport and installation costs as well as maintenance costs. Depth has the same effect on these cost groups. Given the little difference between these factors





in the locations, the alternatives do not suffer large variations among them either in terms of costs. The alternative with a lower life cycle cost is A Guarda-Baiona 1 with 1.223.896.612 €.

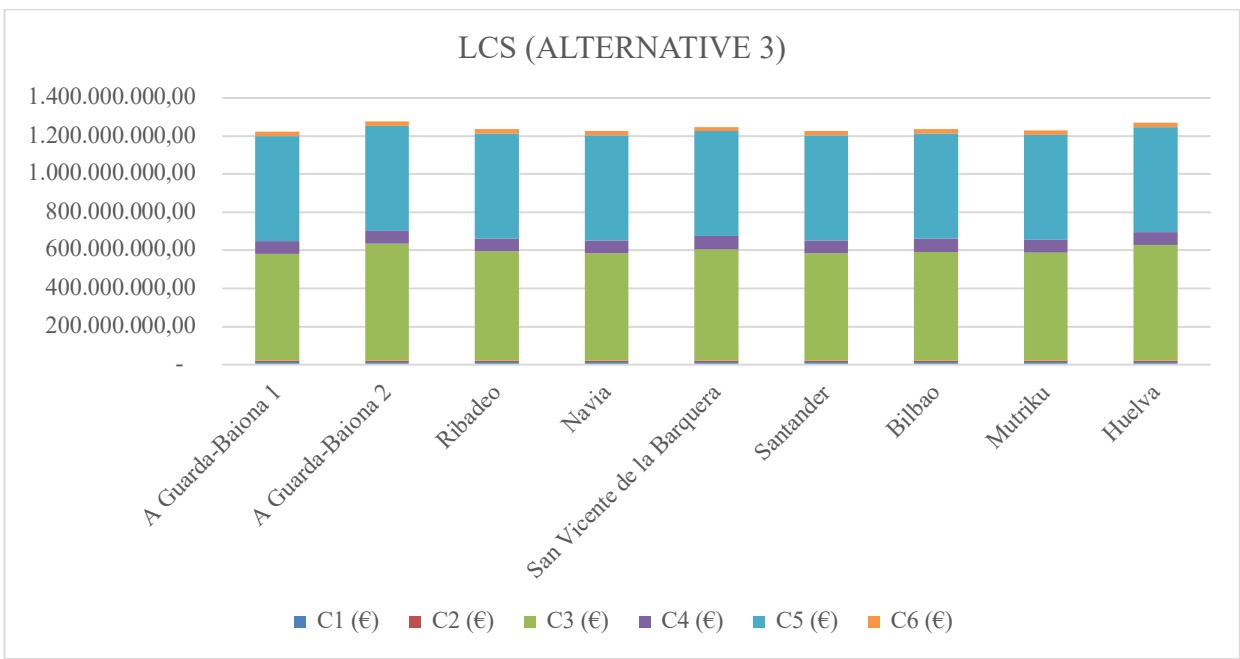


**Figure 5.** Life-cycle cost of the location

Other input values are constant for all locations as they characterize the floating offshore wind farm, such as its size, quantity and power of the turbines, distance between turbines, turbine model, floating technology, etc. These data are mainly entered 135 in the manufacturing cost equations. That is why cost group C3 is very similar for all locations. The maintenance costs may vary depending on the distance to the coast. The greatest inequality between costs is A Guarda-Baiona 1 and A Guarda-Baiona 2 (53.865.063 €), which coincides with the greatest difference between the distance to the coast from the windfarm, 8 and 35 km respectively.





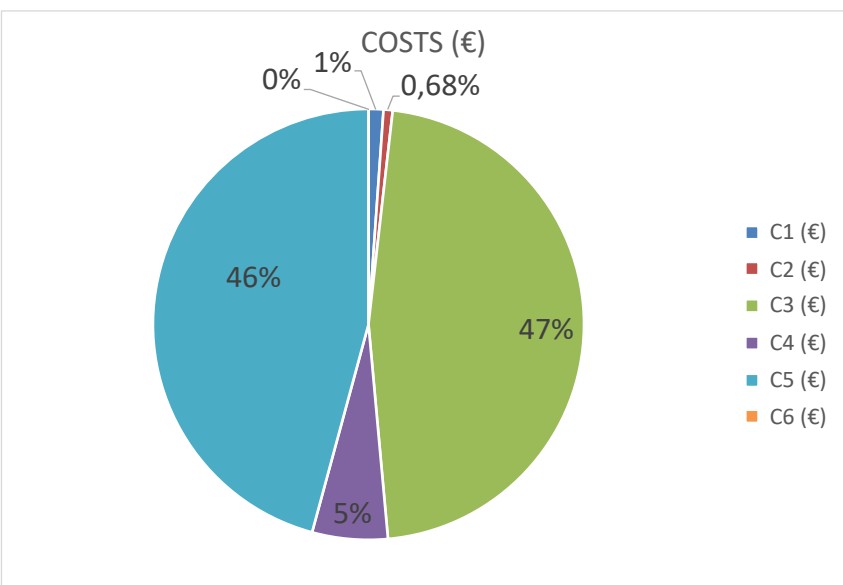


**Figure 6: Life Cycle Cost of Huelva.**

Figure 6 shows the cost percentages of the life cycle phases of a floating offshore wind farm in Huelva (Spain). Manufacturing and operation are the most important phases, reaching 46% and 47% respectively.

Once the cost analysis has been performed, the energy production is calculated based on the procedure explained in Section

**¡Error! No se encuentra el origen de la referencia.**. In this way, the information necessary to calculate the LCOE is available. Figure 7 shows the LCOE results for each location.

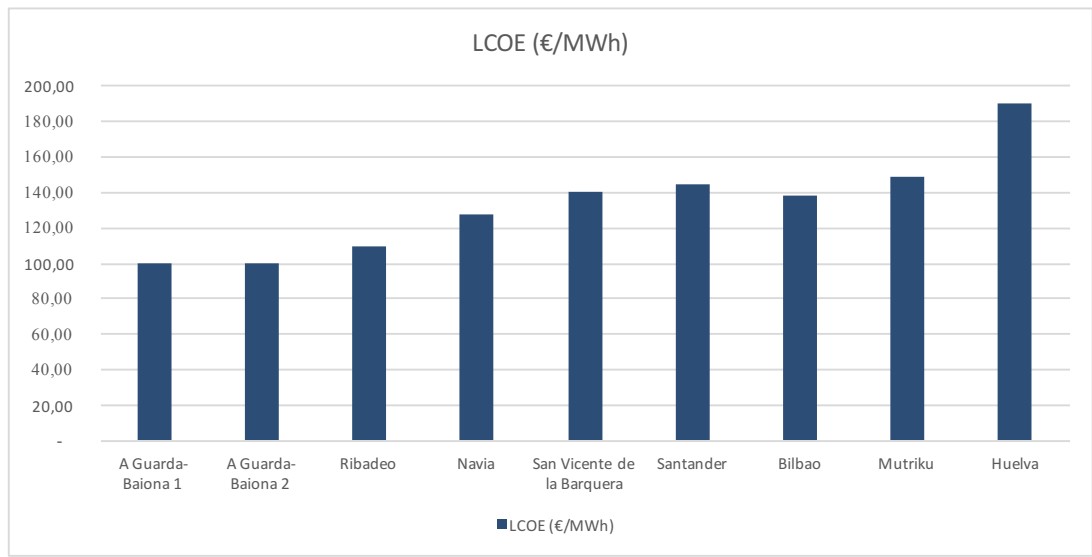

**Figure 7: LCOE results**





Among the locations analyzed, A Guarda-Baiona 1, Guarda-Baiona 2 and Ribadeo have shown the best LCOE values. The lowest LCOE of all is A Guarda-Baiona 1 with € 99.73 / MWh. Huelva has been the locations with the highest LCOE, exceeding 180 € / MWh.

This comparison would allow us to select the best location for an offshore wind farm, but it does not take into account other relevant factors such as electricity tariff and financial rates.

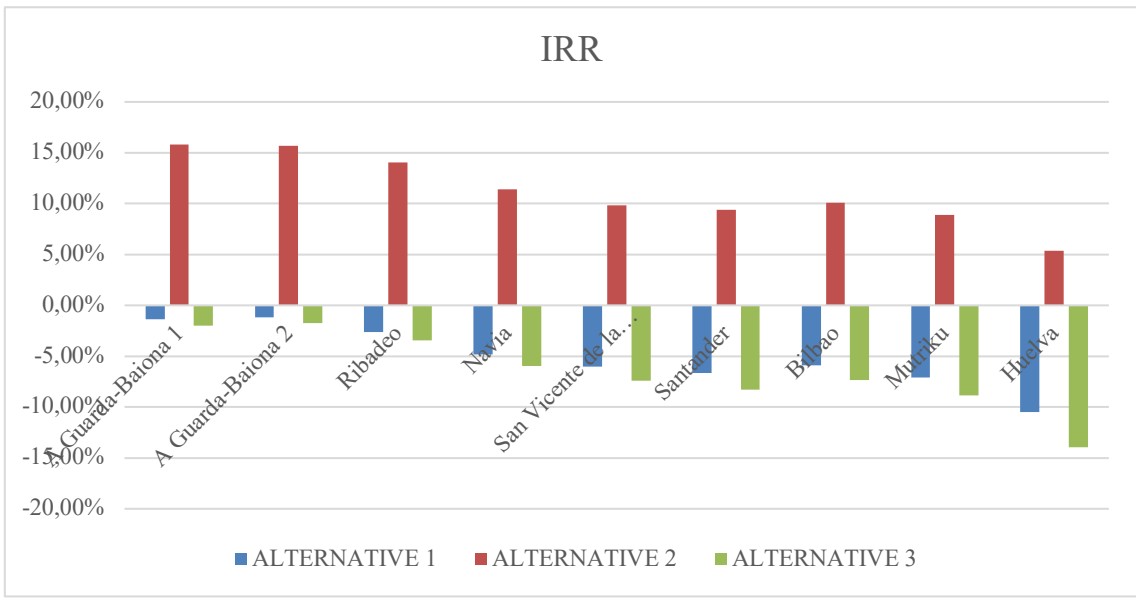


**Figure 8: IRR results.**



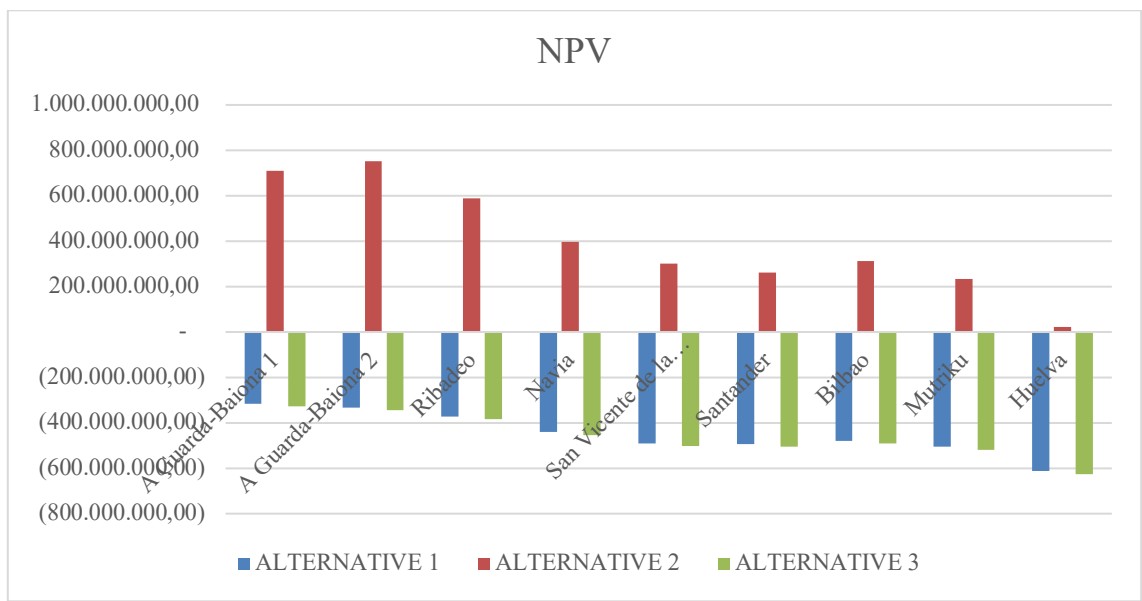

**Figure 9: NPV results.**

In both alternatives, the electricity rate considered is low and causes both the NPV (Figure 9) and the IRR (Figure 8) to offer negative values for all study locations. As in the case of the LCOE, the best location is A Guarda-Baiona 1, with an IRR of -2% considering dismantling costs

The results also show the great importance of the electricity tariff. Figure 9 shows NPV results for all the locations and the
alternatives, and how it increases when using a high electric tariff, reaching 750.000.000 € for A Guarda-Baiona 2.

Even if A Guarda-Baiona 2 is the one that showed a higher life cycle cost, its NPV and IRR results are good due to its wind resource.

**5 Conclusions**

The purpose of this paper has been to present a holistic approach to assess the feasibility of floating offshore wind farm in a
life cycle perspective. The methodology proposed analyses the Net Present Value, the Internal Rate of Return and the Levelized Cost of Energy of the farm.

The case study is built based on a disruptive floating spar-type platform called TELWIND, to be implemented in Spain, located in the Atlantic region of Europe.

Results indicate how important these parameters are in economic terms and shows the pathways to reduce the costs of this
type of infrastructure.

The applied methodology has allowed us to select the best location of all those proposed in the case study: A Guarda Baiona 1. The wind resource and the location characteristics make it the best, with a LCOE of € 100 / MWh. This location also has the lowest LCS, which shows a relationship between both calculations.



Huelva shows the worst LCS results and also the worst economic parameters (IRR of -13.96% in alternative 1). However, the
case of Guarda Baiona 2 also has high LCS values, but the economic results are favorable. Which means that the wind resource
plays a very important role in the results of economic feasibility of a floating offshore wind farms.

Furthermore, the methodology proposed allows the selection of the best region where a floating offshore wind farm can be
installed. Finally, this study can be useful for Governments and relevant authorities to determine the best location of a floating
offshore wind farm and develop the roadmap of offshore wind in their country.

**Acknowledgment**

This work was performed in the scope of the ARCWIND project (EAPA_344/2016), co-financed by the European Regional
Develop Interreg Atlantic Area Programme.

The authors would like to thank the enterprise Esteyco for the information provided.

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



Union, E.: Acuerdo de París | Acción por el Clima, n.d.

Viaintermedia.com: Eólica - La eólica mundial suma ya 591 GW - Energías Renovables, el periodismo de las energías limpias., n.d.

Wind Europe: Wind energy in Europe in 2018, n.d.