# Peer review of "Pathways to bring the costs down of floating offshore wind farms in the Atlantic Area"

_Wind Energy Science, 2019_

## Short Comment (SC1) · 5 Dec 2019

This study is very interesting because it analyses concrete platforms, which are in development yet. Do you consider that this type of platforms will be competitive in the long term?

---

## Short Comment (SC2) · 5 Dec 2019

RECOGNICED VALUES.-

1.- This work is part of a European project Project Interreg ATLÁNTICA AREA involving up to 15 partners, who have different aspects.

2.- Specifically, this paper studies the real costs of a real telescopic semi-submersible concrete platform (TELWIND$^{®}$ platform), suitable for medium-high depths to those of the future market. In addition, nine locations are studied on the Spanish coast. I understand that this same study can be extended to any other location, only by varying the input data.

3.- The costs are studied in all phases of the project. From the beginning conception

and definition to its final Dismantling, through the design and development, manufacturing, installation and commissioning and exploitation. Collect up to 938 entries, obtained from other phases of the project, made by other European project partners.

4.- It is concluded that the highest difference in maintenance costs (reaching 54 million euros) is due to the distance between the farm and the base port, from where maintenance, repairs and final dismantling are carried out.

5.- The results show that the difference in the values of the Levelized Cost of Energy of the farm (LCOE) can be doubled, depending on the location. This is the case of Huelva (exceeding € 180 / MWh) compared to the Baiona-A Guardia (€ 99.73 / MWh).

6.- The influence of the variation of the electricity tariff, representative of scenarios that have been given or that can be expected, is analyzed to see its effect on economic parameters of farm operation. It is concluded that the operating results depend a lot on the electricity tariff.

7.- It also allows you to quickly study the influence of other park parameters such as the number of generators in the park, size, distance between turbines, etc.

8.- In summary, I think it represents a useful tool to study different alternatives. It provides valuable economic information that can help the governments of each country to establish their policy of future investments in offshore wind power for renewable electricity generation, objective of the Paris agreement of 2015 (COP21) and possibly enhanced in the commitments adopted in the one in Madrid (COP25 - December 2019).

In summary, I recommend the publication of this work.

MINOR FIXES

1.- In line 145 reference error appears about section of the work. It must be corrected.

2.- In graphs 8 and 9 (that I attached), the titles of locations are superimposed on the

scale of the vertical axis, making it difficult to read. It must be corrected.

[Figure]

[Figure]

155

**Figure 8: IRR results.**

**Fig. 1.**

[Figure]

160 **Figure 9: NPV results.**

**Fig. 2.**

---

## Author Comment (AC2) · 5 Dec 2019

Thank you for considering our article.

Regarding your question, we think that this type of platforms will be competitive in the future due to the nature of their material (concrete has good offshore conditions) and their costs.

---

## Short Comment (SC3) · 9 Dec 2019

Could you send me the two modified figures (n° 8 and 9)

---

## Short Comment (SC4) · 9 Dec 2019

My last comment today is because in the file wes-2019-73-AC1-supplement.pdf I see that figure 8 is corrected but not # 9
* * *

---

## Short Comment (SC5) · 10 Dec 2019

I Agree. Everything is fine in the version wes-2019-73-AC3-supplement.pdf

---

## Referee Comment (RC1) · Anonymous Referee #1 · 23 Mar 2020

In the manuscript, the authors use the model presented in Castro-Santos and Diaz-Casas, 2014) to analyze the Net Present Value, the Internal Rate of Return, the Payback Period and the Levelized Cost of Energy of a 200 MW floating wind farm in 9 potential locations within three electric tariff scenarios. The analyzed wind farm is composed of a novel spar-type concrete floating wind turbines. Some simple discussions are made on base of the analyzed values.

It is of interest to encourage publications for data of costs in different phases of realistic floating wind farms. However, to be published in Wind Energy Science, the manuscript must be organized as a scientific paper with highlight on scientific significance and quality.

This reviewer believe that the comments in following must be appropriately addressed

before the manuscript could be considered for publishing as a journal paper.

1. The cost estimation model needs to be appropriately validated (e.g. value of each input may need to be given, explained and discussed to show the selection of the value is correct of reasonable, validation for the model is needed to show that all the critical aspects with respect to the cost have been appropriately addressed without error or unreasonable uncertainty, or at least some work is needed to show how good the cost estimation model is).

2.The authors need to clearly address what is the new scientific contribution of the manuscript. The methodology implemented in the cost estimation model has already been published. The very simple discussions with respect to values, e.g. NPV, IRR and LCOE, cannot support the conclusions made by the authors that the manuscript shows the pathways to reduce the costs. In fact, the manuscript presents very limited work with respect to the title of the manuscript (Pathways to bring the costs down of floating offshore wind farms in the Atlantic Area). In addition, due to lack of essential details, the method cannot be repeated by others for selecting locations of wind farms.

3. The discussions with respect to the cost estimation are not comprehensive and lack insight. For example, even without carrying out an analysis, it should be known that the the maintenance cost will increase with increase of the distance to shore. The authors need to highlight scientific values on base of in-deep analysis with respect to the NPV, IRR, LCOE and/or other relevant values and have a good organisation to make the analysis convinced.

In summary, this reviewer feels that the current version of the manuscript sounds like a business report rather than a scientific paper. This reviewer would like to encourage comprehensive publications with respect to cost estimation and methods for reducing floating wind turbine costs. However, this reviewer cannot support for publishing the manuscript in its current version due to the comments mentioned in above.

---

## Author Comment (AC5) · 23 Mar 2020

In the manuscript, the authors use the model presented in Castro-Santos and Diaz-Casas, 2014) to analyze the Net Present Value, the Internal Rate of Return, the Payback Period and the Levelized Cost of Energy of a 200 MW floating wind farm in 9 potential locations within three electric tariff scenarios. The analyzed wind farm is composed of a novel spar-type concrete floating wind turbines. Some simple discussions are made on base of the analyzed values. It is of interest to encourage publications for data of costs in different phases of realistic floating wind farms. However, to be published in Wind Energy Science, the manuscript must be organized as a scientific paper with highlight on scientific significance and quality. This reviewer believe that the comments in following must be appropriately addressed before the manuscript could

be considered for publishing as a journal paper. 1. The cost estimation model needs to be appropriately validated (e.g. value of each input may need to be given, explained and discussed to show the selection of the value is correct of reasonable, validation for the model is needed to show that all the critical aspects with respect to the cost have been appropriately addressed without error or unreasonable uncertainty, or at least some work is needed to show how good the cost estimation model is).

We can not give more information about the method because the enterprise does not give us permission.

2.The authors need to clearly address what is the new scientific contribution of the manuscript. The methodology implemented in the cost estimation model has already been published. The very simple discussions with respect to values, e.g. NPV, IRR and LCOE, cannot support the conclusions made by the authors that the manuscript shows the pathways to reduce the costs. In fact, the manuscript presents very limited work with respect to the title of the manuscript (Pathways to bring the costs down of floating offshore wind farms in the Atlantic Area). In addition, due to lack of essential details, the method cannot be repeated by others for selecting locations of wind farms.

The method is based on previous publications, but it is different because this method is only for concrete platforms and the previous publications are calculated for steel platforms. We can not give more information about the method because the enterprise does not give us permission.

3. The discussions with respect to the cost estimation are not comprehensive and lack insight. For example, even without carrying out an analysis, it should be known that the the maintenance cost will increase with increase of the distance to shore. The authors need to highlight scientific values on base of in-deep analysis with respect to the NPV, IRR, LCOE and/or other relevant values and have a good organisation to make the analysis convinced. In summary, this reviewer feels that the current version of the manuscript sounds like a business report rather than a scientific paper. This

reviewer would like to encourage comprehensive publications with respect to cost estimation and methods for reducing floating wind turbine costs. However, this reviewer cannot support for publishing the manuscript in its current version due to the comments mentioned in above.

We consider that these modifications do not make sense.

---

## Referee Comment (RC2) · Anonymous Referee #2 · 9 Apr 2020

In this paper, LCOE of a floating wind farm based on the concept of TELWIND was performed. The paper requires a major revision.

The language of the paper should be improved. The authors should also use the proper terms in the area of offshore wind.

The paper is too brief and lacks many details in particular on the cost analysis.

It lacks of the details about the TELWIND concept. It is not clear to me whether an engineering design was made so that the structural details are determined and used for the estimation of fabrication cost. The dimension of the concrete platform and mooring system were not given.

On page 3, when calculating the power production, the wind speed distribution and the

power curve were used. However, no detailed values of the distribution parameters and the power curve of the 10MW wind turbine were given. How about the wake effect in the floating wind farm and would it reduce the total power production as compared to the sum of the individuals?

The life cycle cost model was introduced on pages 3-4. However, it is very difficult to understand what input parameters are used and how the unit cost for example for fabrication was obtained. It is also not clear whether the operation and maintenance costs are included or not.

Comparison of the LCOE was made for different locations. But what are the assumptions made behind the analysis. Did the authors re-design their floating wind turbines according to the local environmental data? It is important to compare the power output and the cost of the floating wind turbine separately to see the effect.

---

## Author Comment (AC6) · 10 Apr 2020

We consider that these modifications do not make sense.